# Systems Thinking to Understand National Well-Being from a Human Capital Perspective

**Sibel Eker \*** 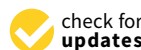 **and Leena Ilmola-Sheppard**

International Institute for Applied Systems Analysis (IIASA), Laxenburg 2361, Austria; ilmola@iiasa.ac.at
\* Correspondence: eker@iiasa.ac.at

**Abstract:** Well-being has become an important policy goal to replace gross domestic product (GDP) as an indicator of national progress. Several multidimensional metrics and indicators of well-being have been developed mostly based on the four-capital model that includes natural, economic, human and social capital. These multidimensional measures of well-being, however, are highly categorical and lack a systems perspective that focuses on underlying mechanisms of the metrics and the interconnections between them. This study aims at bringing a systems thinking approach to understanding and measuring national well-being, particularly from a human capital perspective. For this purpose, we employ a qualitative systems mapping approach and identify the direct or indirect relationships between the well-being indicators related to human capital. The results show that the human capital system is governed by several reinforcing feedback loops through economic progress, health and life expectancy, which gives a central role to human capital to enhance well-being. There are balancing loops, however, that may have adverse effects on human capital formation and well-being, for instance through migration and ageing. Future studies can focus on the other three subsystems in the four-capital model, and on quantifying the relationships between different dimensions of well-being.

**Keywords:** well-being; systems thinking; causal loop diagrams; conceptual modelling; better life index

## 1. Introduction

Well-being has recently become a focal point in the policy agenda of several national governments due in particular to its multidimensional outlook on human welfare and alignment with the United Nations (UN) sustainable development goals. Well-being, in general, refers to the state of feeling or being happy, healthy or prosperous [1]. Earlier philosophical definitions of well-being were based on welfarism, following a utilitarian view on maximizing the pleasant and minimizing the unpleasant [2]. This view was manifested in measuring national well-being in macro-economic terms, such as GDP, which has been the measure of national progress for several decades. However, the inadequacy of GDP to measure national progress and well-being has been increasingly acknowledged [3], and alternative indices based on subjective well-being or multidimensional statistical indices have been proposed [4]. The examples of such multi-dimensional measures start with the UN Human Development Index (HDI) that combines life expectancy and education level in addition to GDP [5], the 10-dimensional well-being measure of the UK's Office for National Statistics [6], and the Organization for Economic Co-operation and Development (OECD) Better Life Index (BLI) that combines a wide variety of metrics from economy to housing and health [7].

Alternative approaches to GDP refer to freedom and capabilities as the main sources of well-being so that individuals can realize their potential [8–11]. Following this capability-based view, the statistical measures of national well-being have become multi-dimensional and led to the four capital model,

which is composed of built (economic), human, social, and natural capital as the main determinants of well-being [12]. These four capitals are further decomposed into various measurable indicators in different studies [13,14], such as air pollution as an indicator of natural capital or voter turnout as an indicator of social capital.

These multidimensional measures of well-being, however, are highly categorical. They identify and list various material and non-material aspects of well-being, yet they do not strongly consider the relationships between these aspects, or the underlying mechanisms that lead to co-development of various well-being indicators. Existing studies increasingly show statistical dependencies between different dimensions and indicators of national well-being [15]. Therefore, current understanding and measurement of well-being can strongly benefit from systems thinking, i.e., focusing on the interconnections between various dimensions of well-being and their underlying mechanisms. With systems thinking, more harmonious, effective and efficient well-being policies can be designed, which can result in gaining multiple co-benefits or avoiding unintended consequences.

The objective of this paper is to bring a systems thinking approach to understanding and measuring national well-being, particularly from a human capital perspective. For this purpose, we employ a qualitative systems mapping approach and identify the direct or indirect relationships between the OECD well-being indicators related to human capital. We derive these maps from the scientific literature on human capital to synthesize the main themes and their relationships in the available scientific knowledge.

Our choice to focus on human capital has been motivated by the central interlinking role of human capital in the existing well-being measurement frameworks such as UN HDI and OECD Better Life Index, as well its significant potential in achieving sustainable development goals [16]. Human capital is defined as "knowledge, skills, competencies and attributes embodied in individuals that facilitate the creation of personal, social and economic well-being" [17]. Human capital has been identified as a significant driver of economic growth [18], and increasingly considered as a key driver of sustainability [19]. Therefore, this study focuses on human capital due to its crucial role within the four-capital model of well-being and extends beyond the economic growth perspective on human capital. It investigates how human capital is formed, maintained and linked to the overall wellbeing of individuals and societies based on the framework of the OECD Better Life Index.

In the remainder of this paper, Section 2 first summarizes the OECD well-being indicators (Better Life Index) and motivates the choice of the OECD framework for this study. It then describes the mapping method followed to derive qualitative systems maps of human capital. Section 3 presents the results of mapping with a focus on the feedback mechanisms that explore the relationships between educational attainment, economic growth and health. Section 4 highlights the insights from these maps and their policy implications, and then discusses the limitations of this study and future research potential.

## 2. Materials and Methods

Defining system boundaries is a major challenge in systems mapping, especially for a complex issue like national well-being that covers a variety of aspects from human psychology to biophysical systems. The scope of this study is determined based on the OECD framework for measuring well-being, also known as the Better Life Index. This choice has been motivated by two factors. Firstly, the OECD well-being indicators are extensive, well-defined and well-quantified, covering not only the economic and tangible aspects of national well-being such as household income or employment, but also intangible aspects based on self-reported data, such as life satisfaction and social support. They enable measuring the performance of a country over time and compared to the others. Secondly, the OECD framework is well known. Many policymakers and researchers are familiar with these indicators, therefore the penetration of systems thinking to research and policymaking on well-being through this framework can be more accessible.

While the scope of this study, i.e., the breadth and depth of the maps we develop, is based on the OECD well-being indicators and their categorization, the mapping adds direct or indirect links between these indicators, hence it brings systems thinking beyond categorical thinking on well-being. This section explains these two pillars of this study, by first summarizing the OECD well-being framework, then describing the systems mapping approach we follow.

*2.1. Organization for Economic Co-operation and Development (OECD) Well-Being Framework*

The OECD indicators of national well-being reflect the capability approach [20], and relate to the outcomes achieved in the two broad domains: material living conditions and quality of life. These two domains are further divided into topics, such as income and wealth, jobs and earnings, or work-life balance, and subjective well-being. Each topic is operationalized in at least one measurable indicator, i.e., a key statistic, such as Household net wealth, Long-term unemployment rate, Working hours and (self-reported) Life satisfaction, as the "Current Well-being" indicators in the upper part of Figure 1 illustrate.

The OECD well-being framework also includes indicators of future well-being. These are called **resources**, since they facilitate the maintenance or enhancement of wellbeing over time. Resources are categorized with a capital-based approach, i.e., the four capital model, since their change over time is attributed to the current stocks of the capitals. Four main groups of capital are considered: natural, economic, human and social capital. Each capital is represented by a set of indicators, often overlapping with the indicators of current wellbeing [21].

For the indicators of future well-being, the OECD follows a stock-flow categorization. The indicators that relate to **stocks** represent a store of a resource for future well-being. The **flow** indicators relate to the factors that increase or deplete the stocks over time. For instance, 'concentration of greenhouse gases in the atmosphere' is a stock indicator of natural capital, while 'greenhouse gas emissions per capita' is a flow indicator that increases it. A third group of future indicators are called **risk factors**, which are the factors that are not directly outflows of stocks, but can lead to the depletion of stocks. This stock-flow conceptualization aligns with the quantitative and qualitative systems modelling approaches such as system dynamics.

Figure 1 visualizes the decomposition of the OECD Better Life Index as a measure of a nation's current well-being and the future well-being indicators based on the four capitals.

The OECD indicators related to human capital cover mostly formal education, skills and health, as listed in Table 1. While the population-level educational attainment levels and skills are formulated as stocks, because they refer to the accumulation of formal education degrees and skills in a society, health-related factors such as obesity and smoking prevalence are considered as the risk factors for future human capital.

**Table 1.** The Organization for Economic Co-operation and Development (OECD) indicators (Better Life Index) related to human capital and their definitions.

| Indicator | Category | Definition |
|---|---|---|
| Educational attainment | Stock | Percentage of people aged 25–64 with at least an upper secondary education |
| Adult skills | Stock | Mean proficiency in literacy and numeracy of the population aged 16–65 |
| Cognitive skills at age 15 | Stock | Mean score for reading, mathematics and science of 15-year-old students |
| Young adult educational attainment | Stock | Percentage of population aged 25–34 who have attained at least an upper secondary education |
| Life expectancy at birth | Stock | Number of years that a newborn can expect to live |
| Smoking prevalence | Risk | Percentage of people aged 15 and over who report smoking every day |
| Obesity prevalence | Risk | Percentage of the population aged 15 and older diagnosed with obesity |
| Long-term unemployment | Risk | Percentage of the labour force unemployed for one year or more |

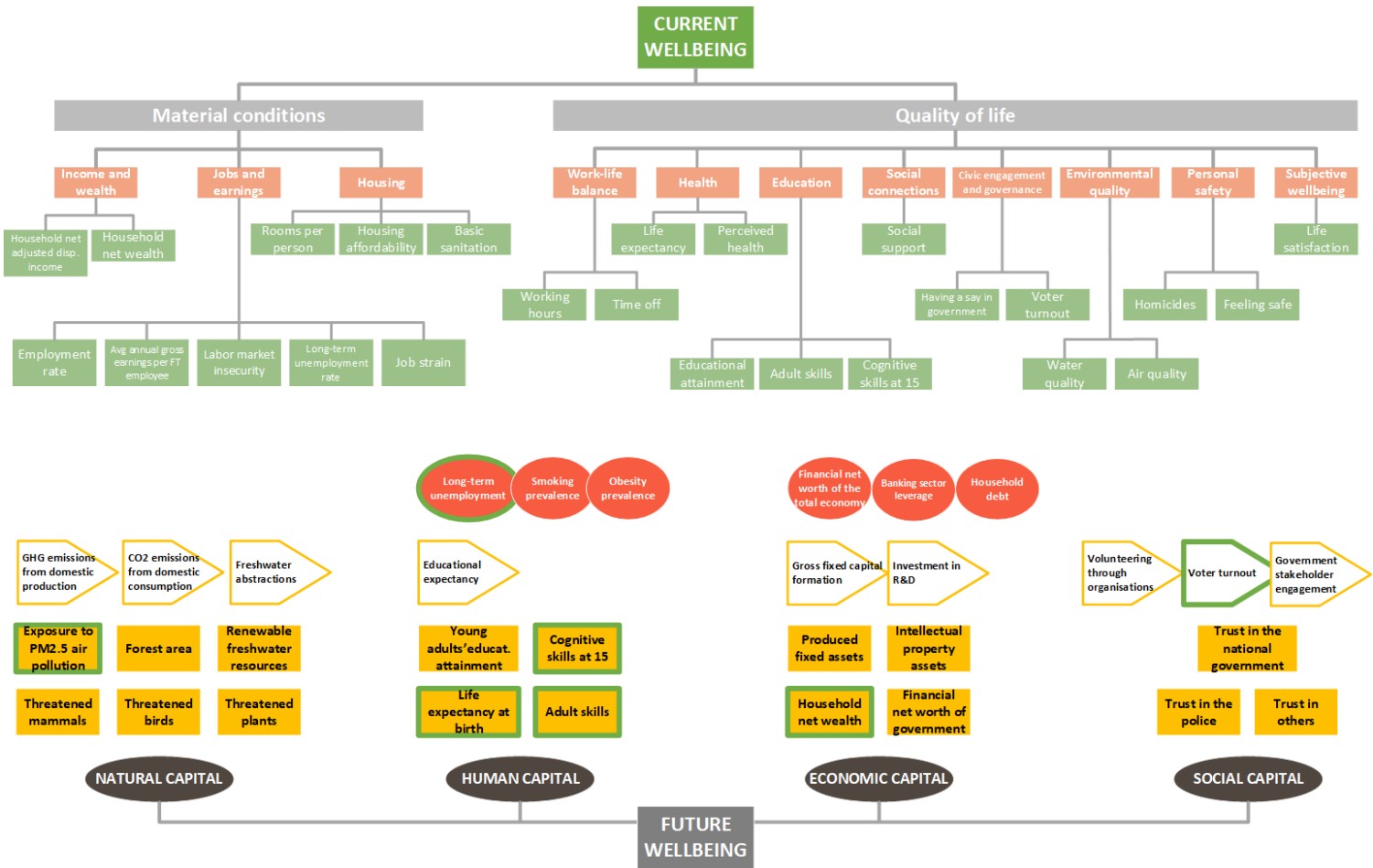

**Figure 1.** Decomposition of the OECD well-being measures framework. The upper part of the figure shows the current well-being indicators, whereas the lower part decomposes the future well-being. The green rectangles depict the current well-being indicators, while yellow boxes refer to the stocks for future well-being. The flow indicators of the four capitals are marked by pentagons and special risks by red ovals. The future indicators with green rim are also an indicator of current well-being, i.e., they are among both future and current well-being indicators.

## 2.2. Systems Mapping Approach

Systems mapping is a primary tool of systems thinking. Based on causal loop diagrams that constitute the basis of system dynamics modelling process [22], a systems map identifies and visualizes the interconnections between system components, as well as the feedback mechanisms formed by those connections. Besides forming the conceptual basis for quantitative models, causal maps depict the multidimensionality and complexity of a problem; help to create mutual understanding of a system by its stakeholders; and aid in propagating the effects of potential interventions to the system qualitatively. Causal systems mapping has been used in a variety of studies, from sustainable consumption [23] to the acceptance of agrifood nanotechnology [24], energy efficiency and well-being in the built environment [25,26], and climate change impacts on mental health [27].

In this study, we chose systems mapping to analyze well-being because it establishes a systems thinking perspective by visually linking the key factors and indicators that drive and define well-being. In other words, systems mapping complements the current understanding of well-being, which is based on compartmentally identified drivers and risks, by adding the interlinkages and non-linearities to those. Systems maps can be derived either in participatory settings with the experts and stakeholders to represent their mental models, or can be elicited by interviewing them, or they can be based on scientific literature. In this study, we have chosen the latter in order to synthesize the academic literature and reflect the available scientific knowledge on the interlinkages between well-being indicators.

The formal methods of deriving causal maps from textual data, whether it is interview transcripts or scientific literature, are based on the grounded theory approach [28]. Based on 'coding', i.e., fracturing and analyzing qualitative data, the grounded theory approach enables a theory to emerge from the data, and links the causal relations underlying this theory explicitly to the data [29,30].

Existing formal coding methods of deriving causal maps from textual data [31–34] have a common framework of five core phases: (i) open coding to identify the themes in the data, (ii) the identification of individual causal relationships, (iii) visualizing these relationships in word-and-arrow diagrams, (iv) generalizing and simplifying these diagrams (axial coding) and (v) recording the links between the final causal map and the data source explicitly in a data source reference table.

The mapping approach followed in this study is based on this framework and specified by six steps, as shown in Figure 2. In Step 1, the literature search is conducted on the Scopus database with key words *human capital* and *well-being*. In other words, we retrieved the articles that include *human capital* and *well-being* in their title, abstract or keywords, published until May 2018, and chose the relevant ones based on their abstract content. Although the concept of human capital could be represented by different keywords such as *human assets* or *human resources*, we chose *human capital* since it prioritizes the studies that relate to the four-capital model. Step 2 is the open coding of the material to identify the themes and main concepts. Step 3 moves the coding further with identification of variables based on the themes and concepts, as well as the relationships between them. By contrast with the original five-phase framework, we compose the data source reference table in this step and use it as an input in Step 4 (Table A1). In other words, Step 3 on axial coding results in the data source reference table which lists the variables and relationships to be aggregated in Step 4. Step 4 aligns the variables identified in the previous step with the OECD indicators and adopts a common terminology. For instance, what the literature refers to as the 'income' of an individual is matched with 'Average annual gross earnings per full time employee' among the OECD indicators. In Step 5, the coding dictionary, i.e., the list of relationships aggregated and matched with the OECD framework, is transformed into causal loop diagrams. This visualization paid special attention to feedback loops and maintained the legend that corresponds to the categorization of current and future well-being indicators as stocks, flows and risks. Step 6 is merging the maps of the four capitals which is to be completed in future studies since this paper focuses only on the human capital.

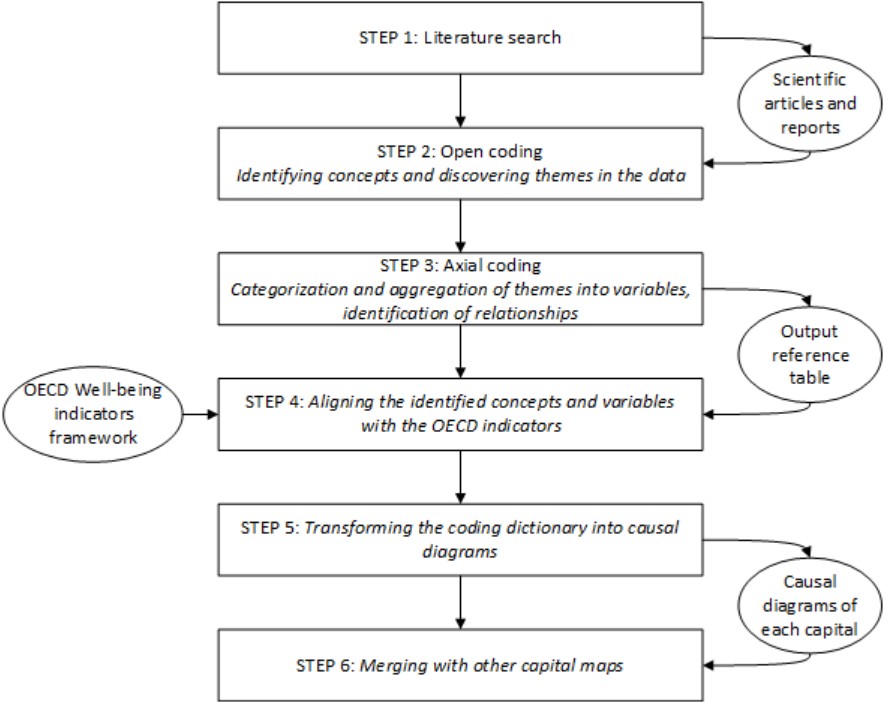

**Figure 2.** Overview of the systems mapping approach.

## 3. Results

As mentioned before, human capital has four main components: (i) knowledge and skills, (ii) experience, (iii) values and norms, (iv) health [18]. The main themes and concepts identified in the literature (Step 2) coincide with these components. Furthermore, since these four components are not separable from an individual, mobilization/migration and societal ageing play an important role in the dynamics of national human capital. Another prevalent theme in the literature is the economic aspect of human capital formation. Formal education, hence knowledge, skills and experience are driven by investments, whether these investments are made by the individuals themselves, their families or governments. The feedback loops governing these investments through generations, i.e., the effect of family on an individual's human capital, are also important drivers of dynamic change at the national-level human capital.

The maps presented in Figures 3 and 4 capture these themes with respect to the relevant OECD indicators. These figures highlight the feedback loops as the drivers of dynamics and non-linearities from two main perspectives, educational attainment and health. These two perspectives represent the main themes that emerged from coding the literature, and the figures include additional factors such as economic growth, experience, and aging in addition to educational attainment and health. While these figures decompose the entire map for an easier tracking of feedback loops, the entire model of the human capital subsystem as a combination of these feedback loops and additional factors is presented in Figure A1.

Throughout these figures, an arrow represents a relationship from the cause variable to the effect variable. A positive (+) sign implies positive causality, meaning that an increase (decrease) in the cause variable would increase (decrease) the effect variable. A negative (-) sign implies negative causality, meaning that an increase (decrease) in the cause variable would decrease (increase) the effect variable. A chain of such relationships forms a feedback loop. A positive (reinforcing) loop implies that a change in any of the variables in this series would reinforce further change in the same direction. A negative (balancing) loop, however, implies that a change in any of the loop variables would balance itself by triggering a change in the opposite direction.

## 3.1. Human Capital from an Educational Attainment Perspective

Figure 3a depicts the positive feedback loops that go through educational attainment (See Table 1). Many of these loops emphasize the positive effect of human capital on national economic growth evidenced by empirical studies [35]. In the first loop (R1), as educational attainment increases, mean years of schooling, i.e., the average education level of a citizen, also increases. Higher education levels lead to increased labour productivity, therefore increased income [36,37]. Increasing income levels (average annual gross earnings) increase the education spending of individuals, either for themselves or for their children [16,18], which then lead to higher enrolment and subsequently higher educational attainment. Considering its definition in Table 1, educational attainment is dependent only on young adult educational attainment, since formal education is followed mostly by the young population, and the education level of the entire adult population changes as the young population's education level changes.

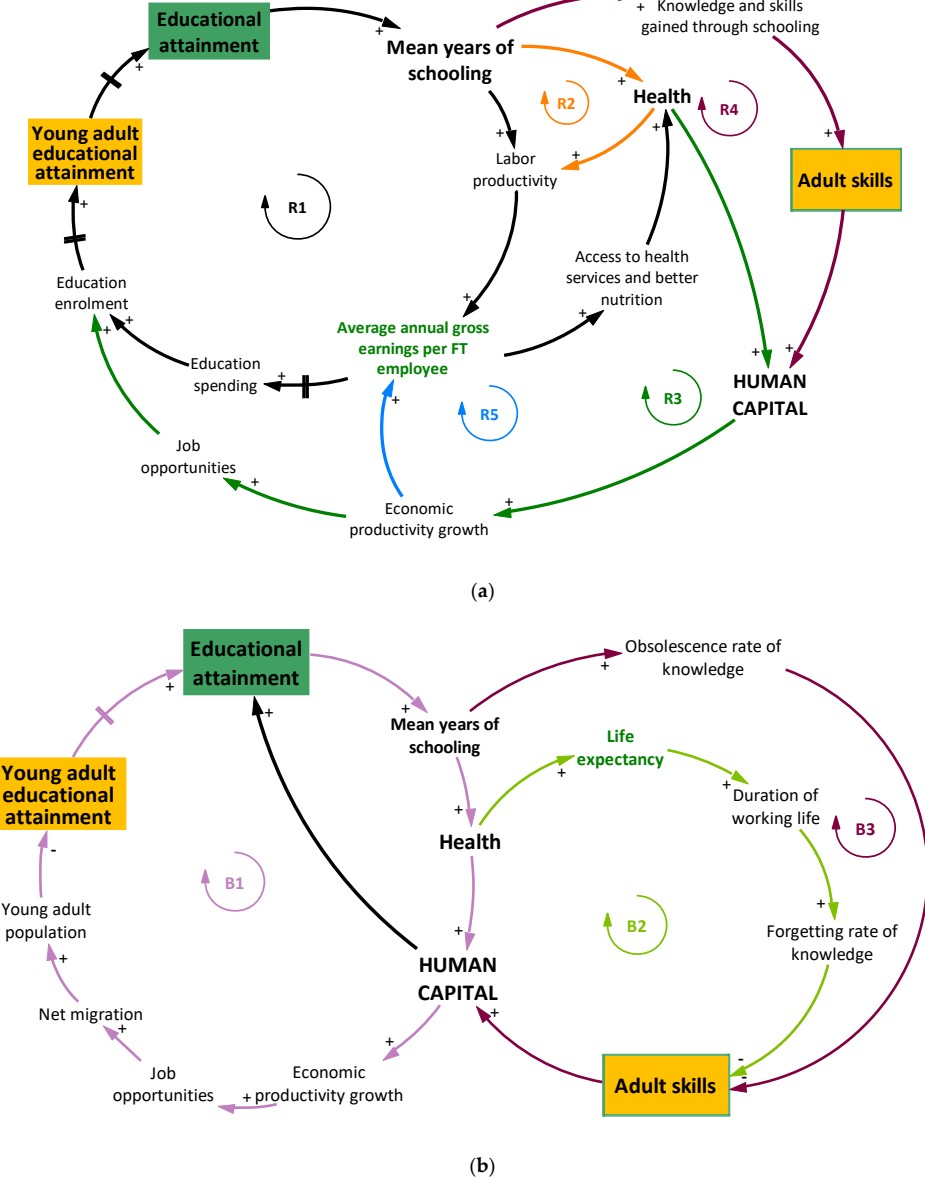

**Figure 3.** Map of the human capital subsystem related to educational attainment. (**a**) Positive (reinforcing) feedback loops, (**b**) negative (balancing) feedback loops. The green boxes refer to the indicators of current well-being, yellow boxes are for future indicators, and yellow boxes with green rim represent both current and future indicators.

**R2** presents an additional path from educational attainment to labour productivity, since the education level of an individual is linked to better health practices, and productivity is affected by health [16,18]. As individuals' health is one of the four components of their human capital, health is an important determinant of human capital also at the country level, which then leads to higher economic productivity growth [38].

Job opportunities created by economic growth increase the potential benefits of education, therefore the enrolment rates [18], leading to higher educational attainment and forming the loop **R3**.

As for **R4**, it depicts the reinforcing effect of educational attainment through skills. Mean years of schooling increase the knowledge and skills gained through formal education, hence the adult skills. Being an important component of national human capital, increasing adult skills lead to further economic growth and close the loop to educational attainment.

The last reinforcing loop (**R5**) is formed by an alternative path from economic productivity growth to education enrolment, because increasing economic growth increases income levels, leading to higher educational spending [16].

These positive feedback loops highlight the importance of human capital for many aspects of well-being, such as income and health. However, they are balanced by negative feedback loops shown in Figure 3b. Therefore, Figure 3a,b should be considered together in order to understand the simultaneous effect of positive and negative loops. The first one (**B1**) reflects a demographic balancing effect of increasing educational attainment. Job opportunities created by increasing educational attainment and economic growth also attract immigration, or lead to lower emigration [18,39]. Considering that migrants are mostly young and mobile people, immigration would increase the young population. Since young adult educational attainment is a fraction of young population, it declines with an increasing population, unless the education level linked to this increase in the population is high, too. It is important to note that this relationship captures only the effect of population increase on the educational attainment metric, and it omits the regulation of immigrant population's education level through government policies because the policy interventions are not in the scope of this study that focuses only on the endogenous system interconnections. However, the relationship between human capital formation and migration is multi-dimensional and more complicated, with positive and negative implications for both the source and destination countries [40,41].

**B2** reflects the effect of aging on human capital. Educational attainment increases health as explained in more detail in Section 3.2, and this leads to a higher life expectancy, which increases the duration of working life for individuals. This implies an aging workforce. Aging leads to higher forgetting rates due to loss of fluid mental abilities such as speed of processing, working memory, long-term memory and short-term memory [42,43]. Forgetting rate of knowledge is one of the two factors that reduce adult skills stock, hence the human capital [37]. The link from human capital to educational attainment is the series of positive links through economic productivity growth, as discussed earlier, that complete the loop. The last balancing loop (**B3**) pinpoints the vulnerability of human capital gained through higher education to technological progress. Most knowledge and skills gained through higher education are those that become obsolete most quickly due to technological progress. A higher means years of schooling implies more higher education graduates in a society, whose specialist knowledge and skills are more vulnerable to technological progress [37]. Therefore, the higher the mean years of schooling, the higher the obsolescence of knowledge and skills. This obsolescence reduces the adult skills stock, which signifies the average value of skills in the population and decreases even though obsolescence affects a small fraction of the adult population (adult skills refer to all technology-relevant skills required in the workforce here, not only the literacy and numeracy rates as measured in the OECD Better Life Index.). This change in adult skills in turn reduces educational attainment through economic growth, which is shortened to the bold black link in the figure.

### 3.2. Human Capital from a Health Perspective

With growing evidence, education reduces adult mortality and increases life expectancy [16,44]. Figure 4 provides a closer look into the health component of human capital through several of the OECD indicators. The two risks for future human capital identified by the OECD, smoking and obesity prevalence, do actually stand in positive feedback loops. The reinforcing loop **R6-wise and healthy** depicts that a high national human capital leads to economic growth, and further educational attainment. Highly educated people follow better health practices [18,45], reducing the smoking and obesity prevalence in the population and leading to better national health, hence higher human capital. This loop implies that these risks for human capital (e.g., smoking and obesity) may not be as high as expected, as long as higher education levels ensure better health practices. Higher income levels induced by economic growth provide more access to better nutrition, reducing obesity prevalence, and more access to health services. These relations form another reinforcing loop, **R7-wealthy and heathy**. It must be noted that the relationship from economic growth to access to better nutrition and health services depends on the distribution of wealth and government policies on this issue. However, since this study focuses on endogenous system relationships, such exogenous policy variables are not included in the map.

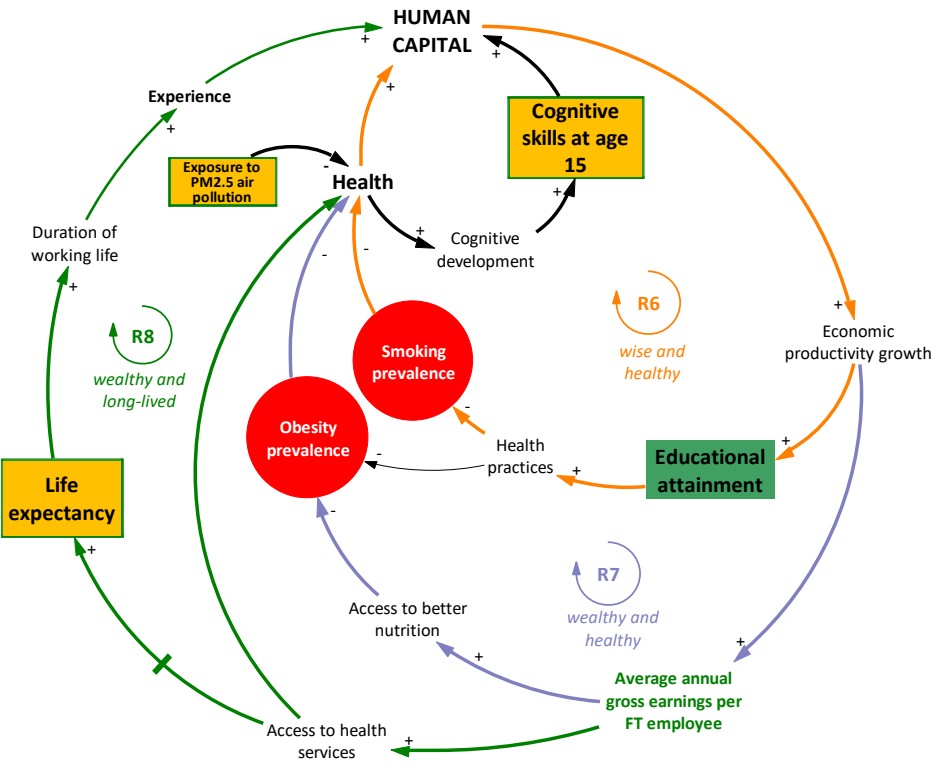

**Figure 4.** Map of the human capital subsystem related to health. The red circles represent the 'risks' for future well-being.

Another health-related reinforcing loop is the **R8-wealthy and long-lived** loop. As human capital increases, the income levels, access to health services, and life expectancy increase, too. This leads to a longer working life, and more experience in the workforce. Being one of the four main components of human capital, more experience leads to a larger human capital stock [18]. It is important to note that the other education-related factors such as health practices impact life expectancy, too, although not visualized in Figure 4, making the education level the most important determinant of life expectancy [46].

Exposure to PM2.5 air pollution is one of the current and future natural capital indicators (Figure 1) and reported to have an important effect on the health and cognitive development of children [16,47].

This link exemplifies the impact of natural capital extending to human capital, since higher levels of air pollution reduce cognitive development and affect cognitive skills at age 15.

Human capital has wider impacts extending beyond those discussed above. The entire map of the human capital subsystem derived in this study (Figure A1) captures two of these additional links. Firstly, education has a well-documented effect on population growth through fertility and infant mortality rates. On the one hand, increasing female education in the young population decreases fertility rates all over the world [48–50], hence impeding population growth. On the other hand, higher education levels of both females and males decrease infant mortality [51], having a positive effect on the young population.

The second wider impact of human capital is on social capital. An increasing level of education is associated with a stronger sense of civic duty, leading to a higher participation in politics [52]. The positive relationship between human capital and democracy has been established also in empirical studies [53,54]. Therefore, higher political participation corresponds to higher voter turnout among the OECD well-being indicators. Still, the interconnections between human capital and other OECD indicators related to social capital should be investigated further.

## 4. Discussion

### 4.1. Policy Implications

This study contributes to understanding and measuring national well-being by initiating a systems thinking perspective on the human capital component of well-being. In addition to categorical thinking that decomposes human capital into various aspects and operational measures, systems thinking focuses on the interconnectedness of these measures. Through this interconnectedness, systems thinking highlights the opportunities for and barriers to enhancing multiple dimensions of well-being simultaneously. It helps to avoid unintended consequences that may arise from non-linearities, i.e., worsening one dimension of well-being while trying to improve another.

The systems maps derived in this study show that human capital creates many opportunities to enhance national well-being, either directly via education levels or through its effects on income and health, because it is governed mostly by positive feedback loops. This means that even a small trigger to increase education levels and human capital propagates through the system, creating many exponentially increasing benefits on well-being. Due to these exponential positive effects, such as decreased mortality, fertility, increased health or increased civic engagement, human capital is considered a key driver of not only national well-being but also sustainable development [16]. However, it must be kept in mind that these loops may run in the opposite direction and create cascading negative effects on human capital, the economy and public health. Therefore, unless their positive functioning is maintained or fostered, these positive feedback loops may be detrimental for national well-being. For instance, if the school systems do not lead to high graduation rates, hence a high educational attainment, this would lead to lower and lower schooling through the economic effects, and would further lower the human capital. Therefore, education policies should pay attention to such wider and longer-term implications of systemic feedbacks.

The balancing loops in the maps highlight the issues that require careful policies to maintain the positive effects of human capital on well-being. Although this mapping study cannot quantify their magnitude, three main risks can be listed based on these balancing loops. Firstly, national human capital creates economic growth and new job opportunities, which attract migration. Migration may cause a decline in the national human capital in return. Secondly, human capital increases life expectancy through many health benefits. Longer working lives, however, lead to ageing of the human capital in return. Lastly, while higher education increases human capital, the knowledge and skills it brings are more susceptible to obsolescence. Therefore, human capital policies should counteract these potential negative effects of migration, aging and specialized higher education. Education programs

for migrants, or life-learning programs to refresh and diversify the capabilities of an ageing or highly specialized workforce could be practical policy options to reduce the impact of such negative feedback.

### 4.2. Limitations and Future Research

Well-being is an immensely broad concept and the systems that determine national well-being are accordingly extensive. Therefore, system delineation is of crucial importance to study well-being. In this study, we chose the indicators and framework of the OECD Better Life Index (BLI) to determine the main subsystems and components to be included in the system maps. BLI is the most extensive and multi-dimensional index, including intangible aspects of well-being such as self-reported life satisfaction in addition to many tangible aspects. Still, it does not include measures about the underlying mechanisms. For instance, while the indicator "Cognitive skills at 15" represents the quality of the school system, there is no indicator about the capacity of the schools, such as the number and qualifications of teachers. Therefore, future studies can deepen the maps by expanding to specific underlying factors.

This study focused on human capital, which is only one component of the four-capital model used to measure well-being, yet included the BLI indictors and additional variables that relate to the other capitals. For instance, average annual earning per full-time (FT) employee is an indicator in the economic capital subsystem, and closely connected to the human capital subsystem since it determines education spending. In future studies, the map of the human capital subsystem should be merged with those of the other capital maps, using such interface variables, in order to obtain the whole systems view on well-being.

Our approach to mapping the human capital system focused on endogenous relationships among the factors defined by the BLI or emerging from the literature. It did not include exogenous policy variables, interventions or regulations. The resulting map of this study can be used in future policy-oriented research, for instance in participatory settings, to identify where different policy options can be imposed on this system and how their effects would propagate throughout the system on multiple dimensions.

This study adopted a general qualitative approach to identify systemic interactions between the drivers and effects of national human capital, without focusing on any particular country. Future studies can contextualize this approach by focusing on specific countries. For instance, the relative importance of each factor can be quantified based on available national statistics or participatory studies with the experts. Moreover, statistical relations between different dimensions of well-being can be quantified based on global databases.

**Author Contributions:** Conceptualization, S.E. and L.I.-S.; methodology, S.E.; formal analysis, S.E.; writing—original draft preparation, S.E.; writing—review and editing, S.E. and L.I.-S.; visualization, S.E.; project administration, L.I.-S. All authors have read and agreed to the published version of the manuscript.

**Funding:** This research was funded by International Institute for Applied Systems Analysis (IIASA) and its National Member Organizations in Africa, the Americas, Asia, and Europe.

**Conflicts of Interest:** The authors declare no conflict of interest.

## Appendix A

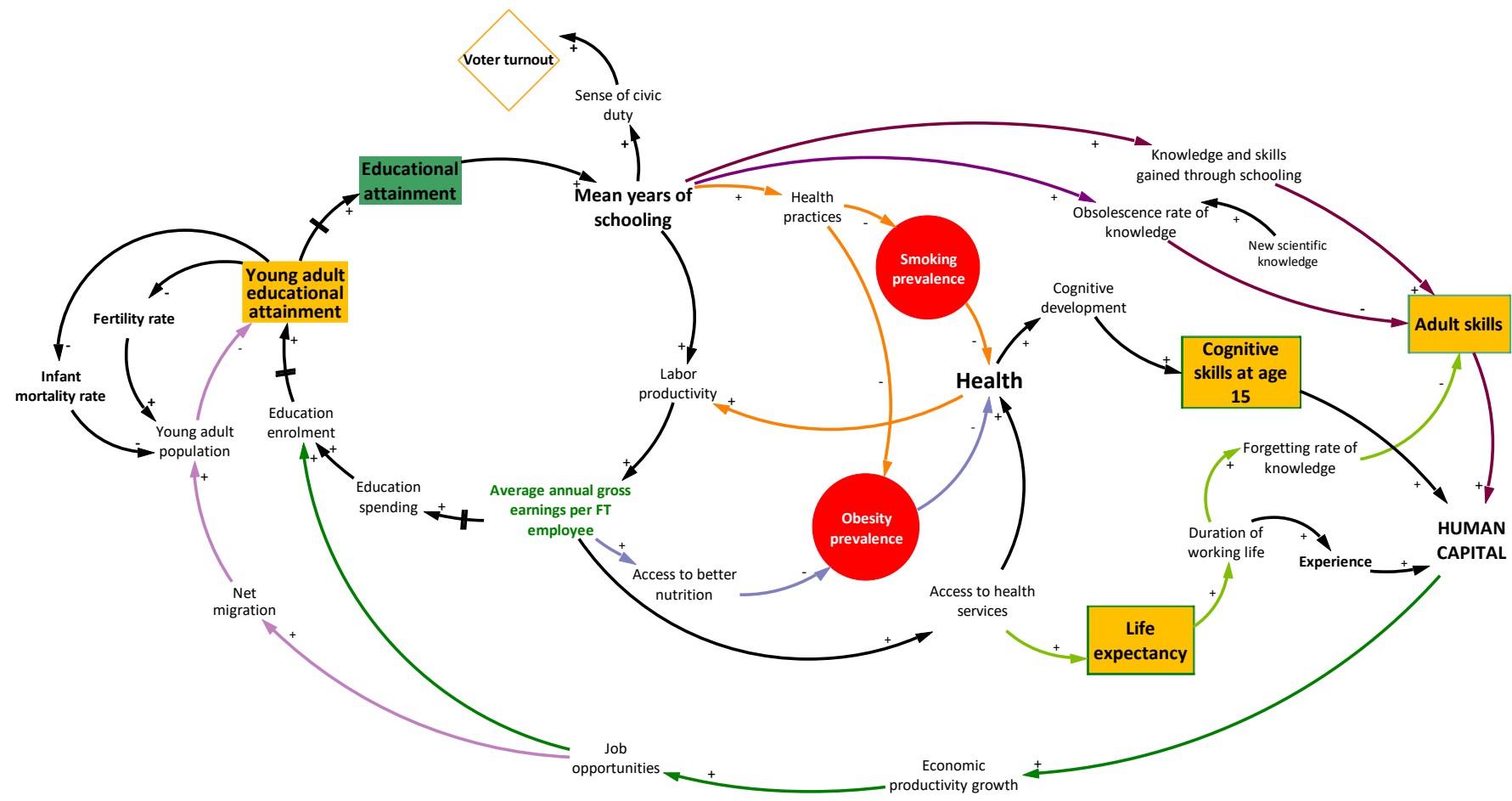

**Figure A1.** The entire map of the human capital subsystem.

**Table A1.** Data source reference table that shows the preliminary coding of scientific articles for the variables and relationships between them. Each row shows a relationship from variable to variable 2. Whether this is a positive or negative relationship is designated by its polarity.

| Variable 1 | Variable 2 | Polarity | Explanation | Reference |
|---|---|---|---|---|
| Education | Income | + | In Step 4, these specific factors and relationships are aggregated into 'Education spending' and 'Job opportunities' increasing 'Education enrolment'. Education spending and job opportunities are triggered by 'Average annual gross earnings per full-time (FT) employee' and 'Economic productivity growth', respectively. Becker discusses these relationships with concrete examples from women's enrollment in higher education and its effects on the economy. | Becker. 1994 [18] |
| Benefits from (college) education | Education enrollment | + | | |
| Cost of (college) education | Education enrollment | - | | |
| Incentives for disadvantaged groups | Education enrollment | + | | |
| Income | Benefits from (college) education | + | | |
| Investment in education | Education enrollment | + | | |
| Job opportunities | Benefits from (college) education | + | | |
| Experience | Income | + | Experience, as a component of an individual's human capital, increases income and decreases job changes. In Step 4, job changes are not included in the final systems map of this study, while the effect of experience on income is through a longer chain. | |
| Experience | Job changes | - | | |
| Education | Health | + | Several non-monetary benefits of education are listed by Becker. In this study's final maps, they are included as health benefits, fertility rates, and civic duty. | Michael, 1972 [55], Becker, 1994 [18] |
| Education | Smoking | - | | |
| Education | Propensity to vote | + | | |
| Education | birth control knowledge | + | | |
| Education | Appreciation of art | + | | |
| Parents' spending on education | Investment in education | + | These drivers of enrolment rates are aggregated into the link between income (Average annual gross earnings per FT employee), education spending and education enrolment. | Becker. 1994 [18] |
| Government loans | Investment in education | + | | |
| Parents' income | Parents' spending on education | + | | |
| Number of children (household size) | Parents' spending on education | - | | |
| Education | Adoption of technology | + | Represented by the link between Mean Years of Schooling and Labor productivity. | |
| Adoption of technology | Productivity | + | | |
| Income | Investment in health | + | Represented by the link between Average annual gross earnings per FT employee and Access to health services. | |
| Education | Life expectancy | + | Mean years of schooling (MYS) above age 15 is a better indicator of life expectancy than income. Represented by the link between MYS, Health and life expectancy in Figure 3b. | Lutz and Kebede, 2018 [46] |

**Table A1.** *Cont.*

| Variable 1 | Variable 2 | Polarity | Explanation | Reference |
|---|---|---|---|---|
| Human capital stock in the workforce population | Economic productivity growth | + | Represented by the link between Human capital and Economic productivity growth in Figure 3a. | Stonawski, 2008 [37] |
| Number of owners of human capital | National human capital stock | + | | |
| Experience | Individual human capital | + | We take the Experience, Adult skills, Health and MYS (as a proxy of knowledge) into account to represent these components of Human capital. | Healy and Côté [56], Stonawski [37] |
| Knowledge | | + | | |
| Health | | + | | |
| Personality | | + | | |
| skills | | + | | |
| Adult literacy rates | National human capital stock | + | These are historically considered proxies of human capital, with an output-based and education-based approach. According to the OECD framework, we take Educational attainment and Mean years of schooling as the drivers of national Human capital. | Stonawski [37] |
| School enrolment ratios | | + | | |
| Educational attainment | | + | | |
| Average years of schooling | | + | | |
| Learning during formal education | Individual human capital | + | We aggregate these factors into: i. Adult skills increased by knowledge and skills gained through schooling (Figure 3a) ii. Adult skills decreased by obsolescence (Figure 3b), iii. Adult skills decreased by forgetting (Figure 3b) and iv. Human capital increased by experience (Figure 4) | Stonawski [37] |
| Acquiring knowledge during professional life | | + | | |
| Acquiring experience during professional life | | + | | |
| Obsolescence rate of knowledge | | - | | |
| Forgetting knowledge | | - | | |
| Duration of working life | Participating in life-long learning activities | + | These factors are aggregated into the effect of Life expectancy on the duration of working life, then Experience, then Human capital, since the detailed explanation of different types of skills and mental abilities are too detailed for the scope and purpose of this study. | Fürnkranz-Prskawetz and Lindh [57], Stonawski [37] |
| | Performing actions, duties and work | + | | |
| Experience | Efficiency, competence, productivity | + | | Park, Lautenschlager [42], Stonawski [37] |
| Fluid mental abilities of an individual | Acquiring knowledge during professional life | + | | |
| | Acquiring experience during professional life | + | | |
| Crystallized mental abilities of an individual | Acquiring knowledge during professional life | + | | |
| | Acquiring experience during professional life | + | | |

**Table A1.** *Cont.*

| Variable 1 | Variable 2 | Polarity | Explanation | Reference |
|---|---|---|---|---|
| Technological progress | Obsolescence rate of knowledge | + | Different knowledge and skills depreciate at different paces. It is not possible to look at such differences at an aggregate level. Still, Stonawski assumes that depreciation rates of higher education levels are higher. For instance, the human capital of a tertiary education graduate depreciates 8 times faster than that of a primary education graduates does. This relationship is represented by the links between MYS, Obsolescence rate and Adult skills (Figure 3b). | Park, Shin [58], Stonawski [37] |
| Individual human capital due to formal education | Obsolescence rate of knowledge | + | | |
| Memory retention | Forgetting knowledge | - | Aggregated into the positive relationship between the Duration of working life and Forgetting rate, and the negative relationship between Forgetting and Adult skills (Figure 3b). | Stonawski [37] |
| Duration of working life | Memory retention | - | | |
| Human Capital | Labor quality (short term productivity) | | More recently, further research on economic growth, represented by the so-called "new growth" models (e.g. Lucas, 1988; Romer, 1990a; Barro and Sala-i-Martin, 1995), has argued that investment in human capital does notjust improve labour quality at a point in time, but can also lead to technological progress and innovation, i.e., positive "externalities" that increase the productivity of other factors. Therefore, while MYS increases labor productivity, Human Capital stimulates overall economic productivity growth (Figure 3a). | Boarini et al., 2012 [59] |
| Human Capital | Technological progress and innovation | | | |
| Mean years of schooling | Income | + | Represented through the link between MYS, labor productivity and Average earnings per FT employee. | Lutz [16] |
| Young education level | The speed of economic growth | + | Not only the economic growth but also the speed of growth is affected by human capital. More specifically, education level of the young population and the secondary education attainment level (e.g., not primary education) plays a key role [35]. These specific relationships are, however, left out of the mapping in this study, because the differences between the three education levels (primary, secondary, tertiary) are beyond the scope. | Lutz, Cuaresma [35] |
| Secondary education attainment level | The speed of economic growth | + | | |

**Table A1.** *Cont.*

| Variable 1 | Variable 2 | Polarity | Explanation | Reference |
|---|---|---|---|---|
| Education level | Sense of civic duty | + | Following Lipset 1959 [52], empirical studies find a positive association between education and democracy [53,60], and that education is a significant and robust determinant of democracy. In this study, we mention these relationship as the links between human capital and social capital components of national well-being. However, we do not explicitly include them in the maps. | Lipset [52], Lutz [16] |
| Sense of civic duty | Political interest | + | | |
| Political interest | Political participation | + | | |
| Formal education | Executive functioning and cognitive abilities | + | Experimental, observational and neurological studies show that education increases cognitive skills, abstract cognitive skills, reasoning, problem solving and decision making. These are also at the basis of well-established link between education and health. We represent this by the positive links from MYS to Knowledge and skills. | Brinch and Galloway [61], Lutz [16] |
| Health | Schooling (inflow of the human capital) | + | There is a causal effect from Human capital (stock) to Health and Income, but no simultaneous effect in the opposite direction. Income and health can only influence Schooling (flow), which changes average Human capital with a delay of several decades. Therefore, we represent these relationships with the links from Average annual earnings to education spending, then to education enrolment. (Figure 3a) | Lutz [16] |
| Household income | Schooling (inflow of the human capital) | + | | |
| national economic growth | Schooling (inflow of the human capital) | + | | |
| Human capital (educational attainment) | Health-related behavior | + | | |
| Human capital (educational attainment) | Economic productivity/income | + | | |
| Environmental factors (air pollution) | Cognitive development | - | Exemplifying the effect of depleting natural capital on human capital formation. Represented by the link between Exposure to PM2.5 air pollution, Health and Cognitive skills at age 15 (Figure 4). | Grandjean and Landrigan [47], Lutz [16] |
| Mother's education | Child mortality | - | Both within families and within communities in developing countries, mother's education level is a stronger factor than income that affects child mortality. In this study, represented by a link between total young adult educational attainment (not only female) and mortality rates (Figure A1). | Pamuk, Fuchs [51], Lutz [16] |
| Household income | Child mortality | - | | |

**Table A1.** *Cont.*

| Variable 1 | Variable 2 | Polarity | Explanation | Reference |
|---|---|---|---|---|
| Education level | Mental activities at high ages ~Cognitive capacity | + | These relationships refer to another dimension of positive link between Education and Life expectancy, through higher mental activities in addition to better health practices and access to health services. They are aggregated in the links between MYS, Health and Life expectancy (Figure 3b). | Lutz [16] |
| Mental activities at high ages | Life expectancy | + | | |
| Health | Labor productivity | + | While the Health and Labor productivity link is included directly (Figure 3a), the others are represented by the links between Health, Cognitive development and Cognitive skills at age 15 (Figure 4). | |
| Childhood health | School attendance | + | | |
| Childhood health | Cognitive development | + | | |
| Economic growth | Higher educational attainment | + | These specific drivers of high (tertiary) education are left out since the differentiation of different education levels are not considered in this study. | |
| Household income | Higher education enrollment | + | | |
| Education level | Fertility rate | - | Represented by the link between Young adult educational attainment and Fertility rates (Figure A1) | |

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
