# Peer review of "Systems Thinking to Understand National Well-Being from a Human Capital Perspective"

_sustainability, doi:10.3390/su12051931_

Round 1
Reviewer 1 Report
Review of Systems Thinking to Understand National Well-being from a Human Capital Perspective
A brief summary:
I commend the author for taking on such an important and timely topic from the perspective of systems thinking. I believe this is a very good topic, but I believe the manuscript falls a little short on trying to pull-together so many, interconnected concepts. I believe the paper will benefit from a re-analysis that simplifies the scope and boundary of this area. As it stands, I think the causal loop diagrams need to be re-thought. I believe the models are both too broad, and yet not detailed enough in showing the causality involved.
Broad comments:
I believe the paper needs to be re-thought.
Strengths:
I believe this is a very good topic to pursue. However, more work needs to be done before it is ready for publication.
Weaknesses:
Figure 3 in the manuscript shows a Causal Loop Diagram (CLD) of the Human Capital Subsystem related to Educational Attainment. I believe the causal paths shown in Figure 3a and 3b are weak in a few respects. First, there is just a single causal path (1 variable cause) that seeks to explain the path from: Education Enrollment to Young Adult Educational Attainment to Educational Attainment to Mean Years of Schooling to Knowledge and Skills Gained Through Schooling to Adult Skills. So, essentially there is only 1 variable into Health (it is caused by Mean Years of Schooling) and only 1 variable into Adult Skills (it is caused by Knowledge and Skills Gained Through Schooling). A more complete Causal Loop Diagram might explain multiple causes for a key variable. But, in this case I think that some of the variables are really measures (Means Years of Schooling) while others are the causal variables (Educational Attainment, as a factor or concept). So, I don’t think it is accurate to state that Educational Attainment causes Means Years of Schooling (the latter is a surrogate measure of attainment).
Recommendation for Figure 3a: I believe the diagram could be clearer and simpler if the surrogate variables are not used, but instead, simply keep the key concept variable in the diagram.
If Figure 3a is simplified, it might be possible to combine it with Figure 3b. Since the forces are operating on most of the same variables, the reader could see both the reinforcing feedback and the balancing feedback in the same structure. That should improve the model.
I believe there are some problems with Figure 3b as well. First, the Obsolescence Rate of Knowledge is caused by Means Years of Schooling. I don’t believe this causal connection explains this path very well. Is the author trying to say that – “more schooling leads to more and faster obsolescence of knowledge?” In a similar way, the path between Duration of Working Life appears to lead to higher Forgetting Rate of Knowledge. So, on Balancing loop B2, a higher life expectancy leads to more forgetting and lower Adult Skills? This causal path is not at all clear, and this area might benefit from a Stock and Flow Diagram approach where the level of knowledge can either increase or decrease depending on the dominance of one path over another. (an example is provided below)
In addition, the variable used on the left hand-side of Figure 3b require additional explanation. Specifically, we have Net Migration leading to an increase in Young Adult Population which then leads to a decrease in Young Adult Educational Attainment. I believe the causal loop diagram does not fully explain this relationship. Since migration is a controllable variable, the diagram leaves out the decision factors that influence Net Migration. It is possible to control this variable through a country’s legal system. The net flow of new workers can be equal to, less than, or greater than the number of workers needed in the economy.
In Figure 4, we have a number of variables that operate at a macro-level, some of which are mixed in with social variables that may be caused by numerous other factors. For example, obesity and smoking are no doubt influenced by Access to Better Nutrition and Educational Attainment and Health Practice. However, they are also influenced by societal factors such as government policy and law (which is determined by voter preferences and societal concerns). It seems that the model would benefit from variables that touch upon government or democratic influence. For example, the variable Average Annual Gross Earnings per FT Employee would certainly be determined by Economic Productivity and Growth. But, in a macro sense, earnings in society could be very unequally spread so that even though the average annual gross earnings go up, a vast majority of the population might not have Access to Better Nutrition. Recommendation for Figure 4 The model could be improved by incorporation of Public variables (government) in addition to Private (or free market).
Other, Minor Points:
In Section 2.2, line 129, the author states “Based on causal loop diagrams that constitute the basis of system dynamics modeling [22], ….” Actually, system dynamics is based more on stock and flow modeling. In essence causal loop diagrams (CLD) are good at communicating the important feedback loops that arise from system dynamic models. But, CLD models do have flaws and they are certainly not the basis of system dynamics. One must be careful with using CLD diagramming because they can sometimes mis-convey important relationships.

Author Response
Dear reviewer,
We are thankful for your careful review of our manuscript. Your comments have substantially contributed to improve the paper. In the attached document, we present our response (in italics) point-by-point following your comments. We explain the changes we made in the manuscript, or how we accommodate your comments.
Kind regards,
Authors

Reviewer 2 Report
This is a very well-written paper, reporting on well-designed and competently executed study into an important topic. I congratulate the author on this fine work and the valuable insights generated. I would encourage this to be followed by a deeper study looking at the measures of the underlying mechanism, e.g. the quality of the school system (the author gives this example in the discussion section). Such a deeper study would expose additional causal loops not included in the current map. Fr example, in the total map given in Appendix A, the ‘mean years of schooling’ reinforces the ‘voter turnout’, but ‘voter turnout is not shown to have relevance for change in human capital. This looks like an omission at the current level of analysis: it could be surmised that an increase in the average level of education in the voter population would drive changes in political commitments to improve the infrastructure that support human capital development, for example better or more schools. It would therefore be interesting to see such a next-deeper-level of modelling, and this would enhance the utility and impact of this study. I also hope that the author will develop similar models for the other wellbeing capitals.
Author Response
We are thankful for this positive evaluation of our paper and acknowledging the future research points we have identified. We hope to continue, as well.
Reviewer 3 Report
Attached are the comments

Author Response
Dear Reviewer,
We are thankful for your careful evaluation of our manuscript. Please see our responses to your comments in the attached document, following your original comments and in italics.
Kind regards,
Authors

Round 2
Reviewer 1 Report
In diagram b, showing the balancing feedback loops, I would just caution interpretation in the text to note that when mean years of schooling increase and obsolescence increase leading to adult skills decreasing, ... since this is a stock we cannot say whether it will decrease in an absolute sense ... only that it will be less than it would have been otherwise. I would make note of that in the text.
... I may have missed other CLD connections, but I would apply the same reasoning in every case involving a stock. Just to be clear with the readers.
I think this is an important area by the way.
Author Response
Thanks for pointing out the issue of obsolescence rate and the Adult Skills stock. We have revised the text (Page 8 Lines 268-270) to clarify that obsolescence can reduce the Adult Skills stock since it functions as an outflow. However, we also noted in the footnote that obsolescence reduces technologically-relevant adult skills, not basic literacy and numeracy skills as referred in the OECD definition of this variable. We checked the rest of the manuscript and there is no occurrence of a similar issue.